# The origins of money: Calculation of similarity indexes demonstrates the earliest development of commodity money in prehistoric Central Europe

Maikel H. G. Kuijpers[ID]◉*, Cătălin N. Popa◉

Faculty of Archaeology, Leiden University, Leiden, Netherlands

◉ These authors contributed equally to this work.
* m.h.g.kuijpers@arch.leidenuniv.nl

**Data Availability Statement:** All relevant data are within the manuscript and its Supporting information files.

## Abstract

The origins of money and the formulation of coherent weight and measurement systems are amongst the most significant prehistoric developments of the human intellect. We present a method for detecting perceptible standardization of weights and apply this to 5028 Early Bronze Age rings, ribs, and axe blades from Central Europe. We calculate the degree of uniformity on the basis of psychophysics, and quantify this using similarity indexes. The analysis shows that 70.3% of all rings could not be perceptibly distinguished from a ring weighing 195.5 grams, indicating their suitability as commodity money. Perceptive weight equivalence is demonstrated between rings, and a selection of ribs and axe blades. Co-occurrence of these objects evidences their interchangeability. We further suggest that producing copies of rings led to recognition of weight similarities and the independent emergence of a system of weighing in Central Europe at the end of the Early Bronze Age.

## Introduction

Money is a type of commodity that acts as a means of exchange and is standardized to some degree, visually or in terms of their weight [1, 2]. Archaeology can provide a unique perspective on the development of money and systems of weighing over space and time, but the discipline has difficulties with the identification of objects that functioned either as commodity money or as (balance) weights. Typical statistical approaches are inadequate for dealing with the approximation that characterizes prehistoric weighing [3, 4]. What is needed for archaeology to contribute to the history of metrology and the origins of money are methods for identifying standardization on the basis of perceived similarity. A principal challenge at this point is to take the statistical tools employed to express accuracy, and adjust them in accordance to the findings from psychophysics, so that the ordinal and qualitative measurements of weight estimation by hand and sight are taken on board. In short, prehistoric weight units quite literally need to make sense.

**Funding:** The research was supported by the Talent Programma VICI from the Netherlands Organisation for Scientific Research (NWO https://www.nwo.nl/; Grant number 277-60-001: "Economies of Destruction". The funders had no role in study design, data collection and analysis, decision to publish, or preparation of the manuscript.

**Competing interests:** The authors have declared that no competing interests exist.

## Bronze Age commodities

Commodities are archaeologically defined as socially recognized, alienable objects that appear in large numbers and emphasize similarity [5–9]. Candidates of prehistoric commodity money from the Central European Early Bronze Age are the so-called *Ösenringe* (hereafter rings), *Spangenbarren* (hereafter ribs), and possibly axe blades. Rings and ribs appear in the southern parts of Central Europe: the Danubian region of southern Germany, Lower Austria, and parts of the Czech Republic. Axe blades are typically, but not exclusively, found in central and north-eastern Germany, roughly corresponding to the cultural area of the Únětice. In between is an area of overlap where rings, ribs, and axe blades are regularly found together, primarily the Czech Republic (Moravia and Bohemia), though some mixed hoards also appear in north-eastern Germany and Poland (Fig 1). A third geographical region where these objects are encountered is Southern Scandinavia, though in lesser numbers [10] (S1A Appendix).

It is generally thought that different economies operated in the three regions where rings, ribs, and axe blades occur [10]. In the southern region commodification was strongest and the economy is often interpreted as resembling to some degree a market economy, in which rings and ribs represented wealth [12–15]. The Únětice region is better characterized as prestige-

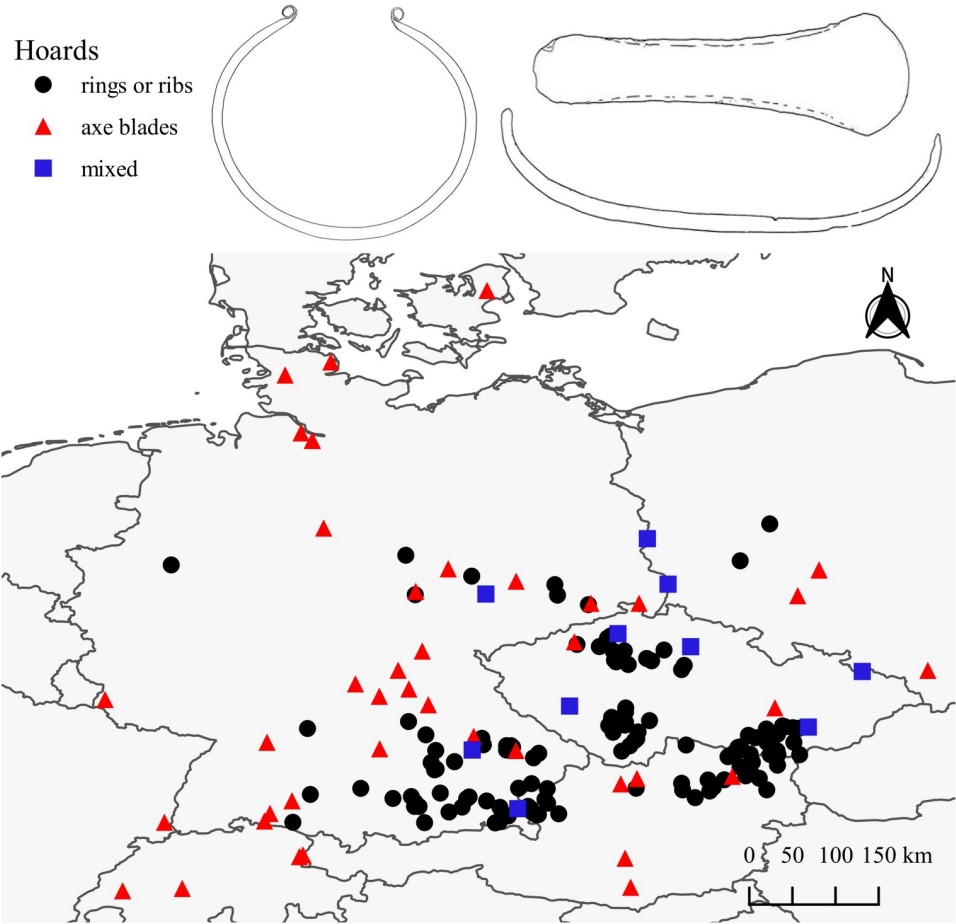

**Fig 1. Map of Central Europe with the hoards used in this study.** Includes drawings of objects after Stein [11] (made with Natural Earth).

good exchange, though some commodification is visible [14]. Scandinavia is different altogether. Here the context of rings suggests gift-exchange, and they mostly appear deposited as single finds in wetlands [10]. A clear contrast between gift and market economies seems hard to sustain, however, and may be little more than a modern distinction [5, 10, 16].

Found in bulk, sometimes in hoards containing multiple hundreds, many of the rings, ribs and axe blades are considered to have no other practical function besides their tentative use as ingots, or rough-outs for further production [8, 17–23]. Moulds, made of clay, stone, or casted directly in sand [24], made serial production possible, which led to some degree of unintentional standardization. However, there are indications that for some types of objects a deliberate effort was made to achieve a specific weight interval, meaning that weight mattered (S1A Appendix). In the case of rings, a standardization between roughly 170 and 220 grams has been hypothesized on the basis of histograms [17–19]. Unclear is how prehistoric people would have recognized this standard.

## Psychophysics of weight discrimination

The practice of weighing may have been far more imprecise than is generally assumed. According to Gyllenbok [25], historically there are three basic ways to measure weight: 1) Through lifting two objects and comparing them, or estimation on sight. 2) Through practicality, i.e. the maximum weight that could be conveniently carried by a human or animal. 3) By means of a weighing apparatus. Each of these have their own level of impreciseness.

In the case of the Central European Early Bronze Age, there is no evidence of a weighing apparatus such as balances. Weighing was a qualitative measurement based on comparative sensory perception with hands and eyes. But humans are known to be "rather 'noisy' measurement instruments" [26]. Prehistoric standardization thus would have been imprecise, and worked based on approximation both in shape and weight. They must have had relatively low precision in terms of modern scientific tolerances [25] and this should be taken into consideration when looking for standardization.

We assume that in the absence of measuring equipment counting must have been the preferred method of quantification, but the counted objects had to be perceptibly similar [25, 27, 28]. Therefore, weight mattered. Weight is crucial for the determination of the value of goods in most economic transactions [3]. Lacking balances, the only way to observe a reasonable degree of uniformity is through sensory perception. We consider objects uniform when they are perceptibly indiscriminate from each other.

Psychophysics offers a methodology through which we can test the perceptible similarity of objects based on weight. This sub-field of cognitive psychology is concerned with the relationships between physical properties of stimuli and perceptual responses to these stimuli [29–31]. The measure used to express people's sensory acuity is the Weber fraction, and denotes the difference in stimulus strength that is just noticeable. As a general rule, the Weber fraction for weight discrimination is 0.1 [32–34]. This means that the difference between, for example, a weight of 100 grams and 105 grams is not noticeable, but between 100 and 111 grams is, since the threshold is at 110 grams (Methods).

## Data

Using the measurement of weight, we employ a methodology based on psychophysics to quantify and operationalize our assumption that weighing in prehistory was a purely sensorial practice, done by hand. The perceptive equivalence is expressed though the calculation of a similarity index (SI), which gives the percentage of objects from a dataset that are perceptibly indistinguishable from a tested object (Methods).

We collected the weights of 6317 objects, of which 5028 were used in the analysis as they were complete and dated to the Early Bronze Age: 2639 rings, 1780 ribs, and 609 axe blades (S2 and S3 Appendices). Axe blades can be divided into Early Bronze I (hereafter: EBA I: 2150–1900 BCE) and Early Bronze Age II (hereafter EBA II: 1900–1700 BCE) on typological grounds [35, 36]. Rings and ribs mostly overlap, though ribs are generally considered a slightly later development [18].

We selected hoards originally containing at least five or more rings and ribs, or at least five axe blades (S1C Appendix). This selection procedure helps identify standardized commodities rather than particular types of rings and axe blades (S1B Appendix). We chose to draw the limit at five because we observed that rings and ribs are found in several instances in bundles of five [17].

## Results

### Rings and ribs

The analysis of the full dataset of rings and ribs revealed the existence of a peak at 193 grams, with a similarity index of 58.6% (Fig 2A). What this means is that when compared to a ring of 193 grams, nearly 60% of all other rings in the database are perceptibly similar in weight. When the dataset was broken into its two main components of rings and ribs, the following picture emerged. The rings, totaling 2639 objects, presented one peak with a similarity index of 70.3% at 195.5 grams (Fig 2B). The 1780 ribs had two peaks. One that corresponded largely with that of the rings, and stood at 186 grams and a similarity index of 44.3%, and a smaller one at 82 grams, having a similarity index of 13.1% (Fig 2C). The separation between the two peaks was estimated using clustering at 135 grams. This value was used to further separate the ribs into a group of heavier and one of lighter objects, which were then analyzed separately. The heavier ribs (n = 1106) revealed the existence of a comparable peak at 185.5 grams, where the similarity index rose to 71.5%. The lighter ribs showed a peak at 81 grams, though the similarity index only reached a maximum of 37.8% (Fig 2D).

### Axe blades

The analysis of all 609 EBA axe blades showed a peak at 285 grams. The peak had a maximum similarity index of only 33.3% (Fig 3A), which is barely over the value expected in the case of randomly distributed data (Fig 4A).

For the EBA I, 208 axe blades were recorded. When analyzed, they showed one extensive peak at just over 206 grams, with a maximum similarity index of 44.2% (Fig 3B). The EBA II gave 401 axe blades. The data showed two peaks, a larger one, with a maximum similarity index of 44.9%, at a weight of 293 grams, and a smaller one, at 180 grams, with a similarity index of 15.2% (Fig 3C), with the limit between the two at 233 grams. Since EBA II axe blades came only from 11 hoards, splitting the dataset further to analyze each peak is unrealistic.

**Combination of rings, ribs and axe blades.**   In order to test whether EBA I axe blades were perceptibly similar in weight to rings and ribs, the datasets of rings, heavy ribs and EBA I axe blades were combined and tested together. Given that there are only 208 EBA I axe blades and 3745 rings and heavy ribs, we took a random sample of the same number from the latter. These were placed together with the axe blades, giving a dataset of 372 objects after the exclusion of outliers. This dataset was then subjected to the similarity calculation. When plotted graphically the results revealed that, although in absolute numbers axe blades were heavier, in terms of weight perception a majority of objects were grouped together in one peak. The top of this peak corresponded to a similarity index of 60.8% at 199 grams (Fig 3D).

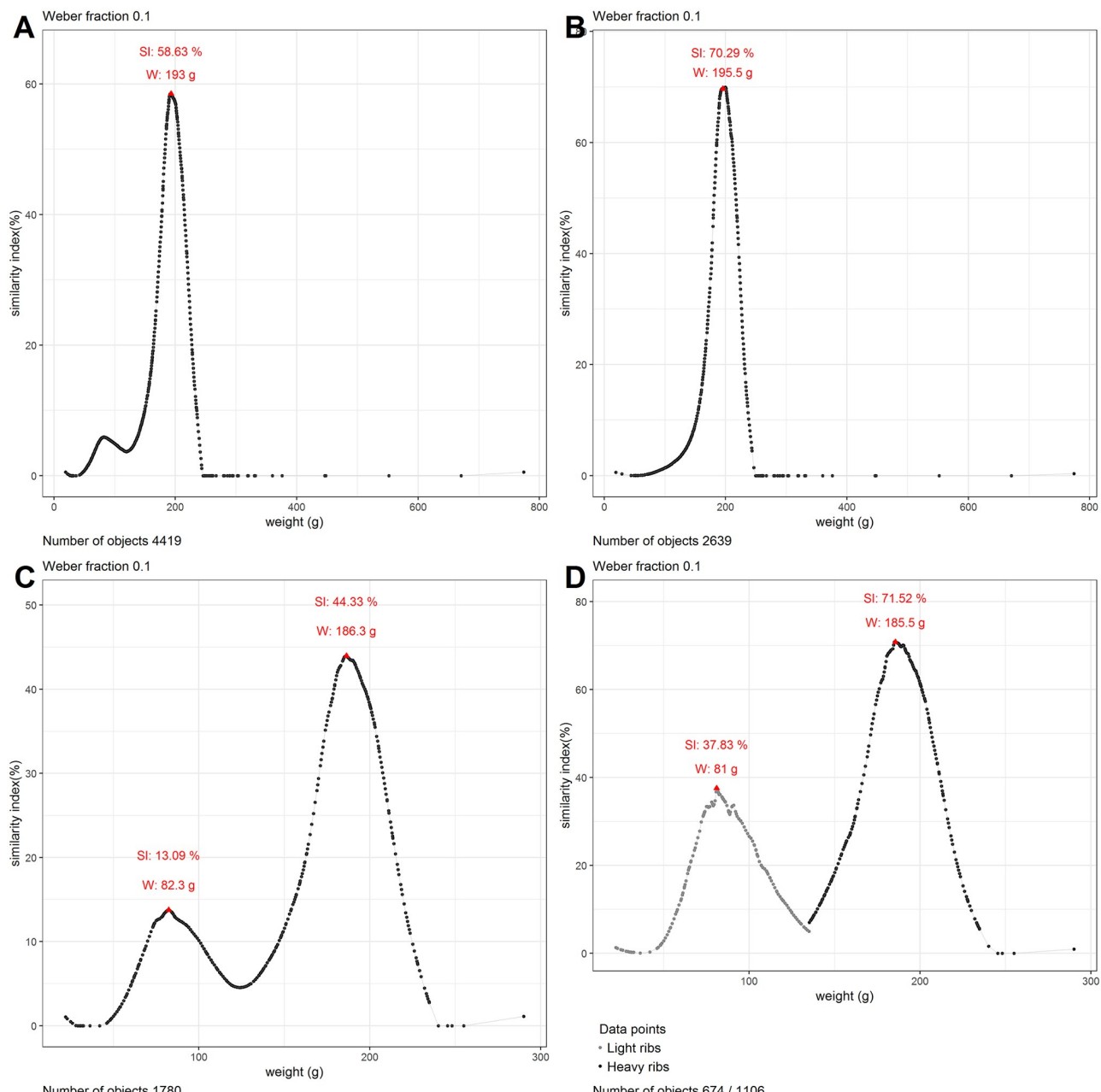

**Fig 2. Similarity graphs for rings and ribs.** The red triangles represent peak tops. Panel A shows the similarity graph for ribs and rings when analyzed together. Panel B shows the similarity graph for rings only. Panel C shows the similarity graph for all ribs. Panel D shows the similarity graph for heavy and light ribs when analyzed separately.

## Analysis

Our findings show that of a total of 2639 rings coming from 113 different hoards, 70.3% (1855 rings) weighed between 176 to 217 grams, making them perceptibly identical to a ring of 195.5 grams. This is interesting as research in psychophysics reports a decrease in accuracy with weights below 200 grams [33, 37]. Rings might have been produced at the lower limit of where differences between them were still easily recognizable. A large similarity was not only calculated

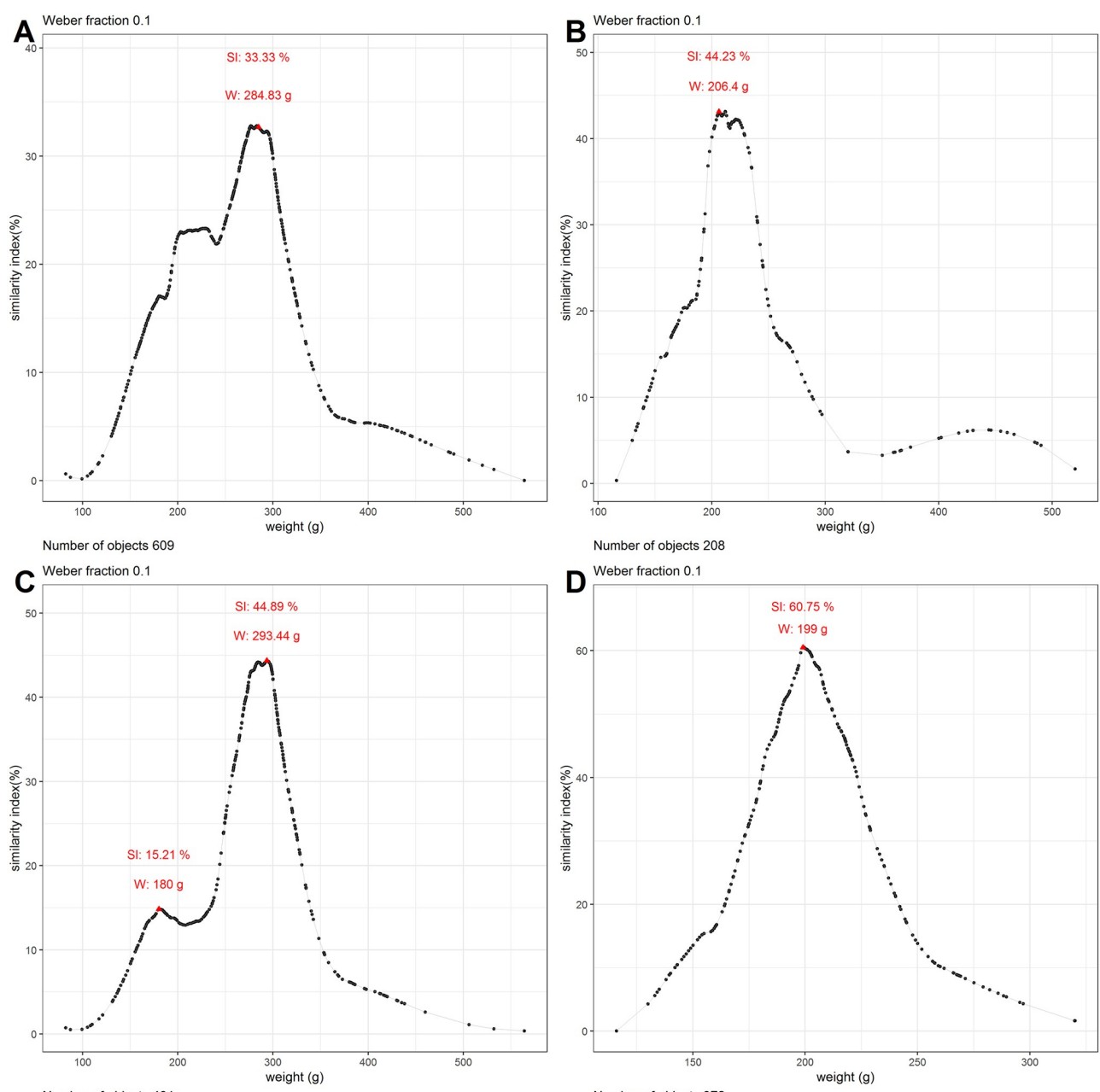

**Fig 3. Similarity graphs for axe blades and combination of rings, heavy ribs and axe blades.** The red triangles represent peak tops. Panel A shows the similarity graph for all EBA axe blades. Panel B shows the similarity graph for EBA I axe blades only. Panel C shows the similarity graph for EBA II blades only. Panel D shows the similarity graph for randomly selected rings, heavy ribs and EBA I axe blades.

for the rings of 195.5 grams, but also for the majority of the dataset: 1724 rings had a similarity index of over 50%. Even the average similarity index of all rings from the dataset was close to 50% and many individual hoards showed a similarity index of 100% (S1A Table in S1 Appendix). When excluding outliers, the similarity of the rings rose to 76.5%, while randomly generated data over the same weight interval remained below a similarity index of 49% (Fig 4C).

Part of the ribs revealed a comparable situation with the rings. Of the 1106 heavy ribs, coming from 13 hoards, 71.5% were perceptibly identical to a rib weighing 185.5 grams, and

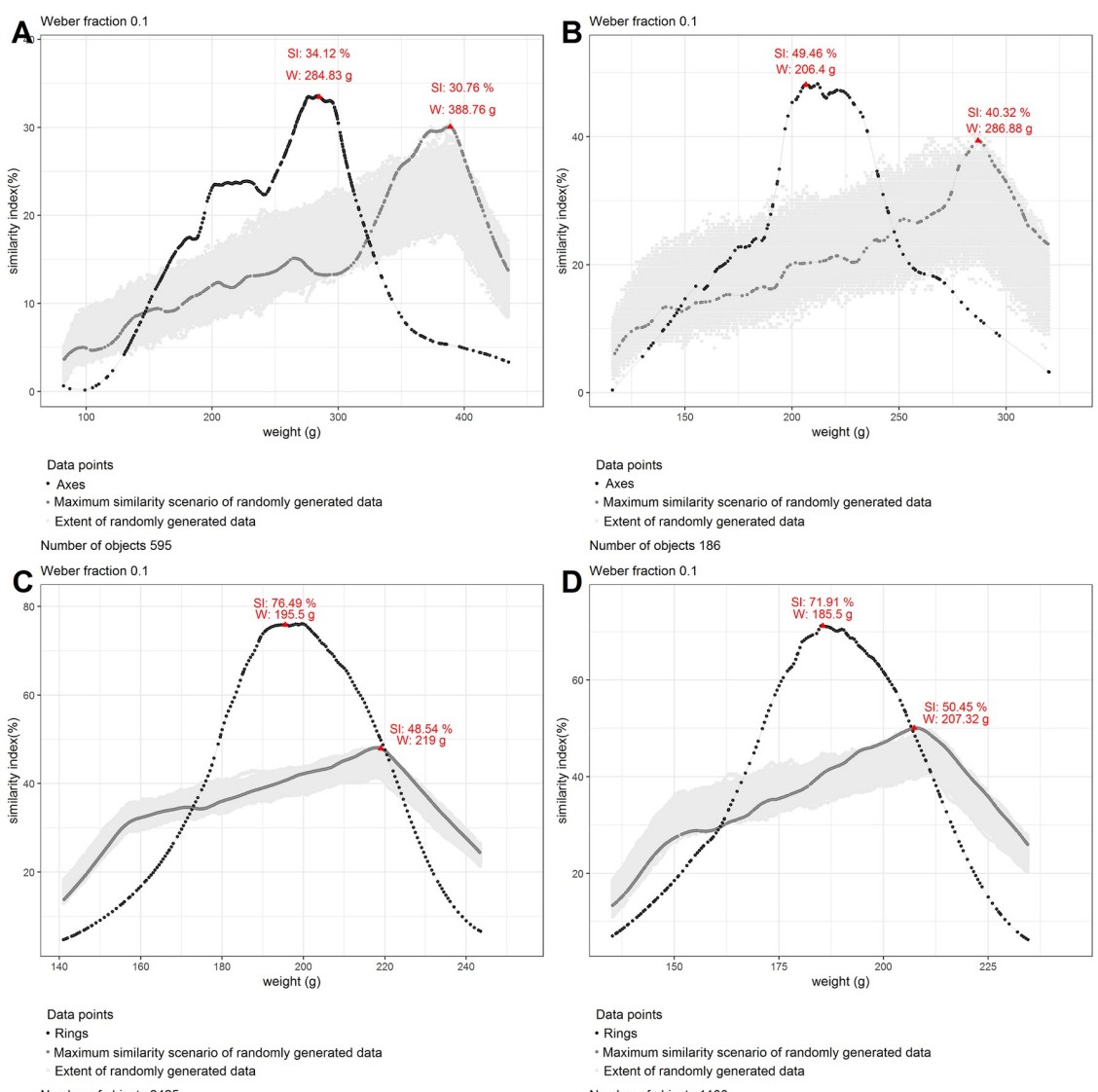

**Fig 4. Similarity graphs for axe blades, rings and ribs with randomly generated data.** The red triangles represent peak tops. Outliers have been excluded. Panel A shows the similarity graph for all EBA axe blades with randomly generated data. Panel B shows the similarity graph for EBA I axe blades with randomly generated data. Panel C shows the similarity graph for rings with randomly generated data. Panel D shows the similarity graph for heavy ribs with randomly generated data.

weighed between 167 and 204 grams. Just like with the rings, a majority of these ribs (n = 741), showed a similarity index of over 50%. Randomly generated data over the same weight interval revealed a maximum similarity that was 20% lower than that calculated for the heavy ribs, when excluding outliers (Fig 4D). The hoards themselves showed a high degree of internal homogeneity in terms of weight, as all of them had an average similarity index of over 60% (S1B Table in S1 Appendix). Ribs therefore, just like rings, show a strong pattern of perceived standardization. Though our analysis shows that the targeted weight was 10 grams smaller than that of the rings, this weight interval could not have been perceived when comparing these objects and thus this standardization overlaps with the rings in what we suggest calling a *perceptive category*. This refers to a recognizable range of sensorial parameters which humans could reliably identify [38, 39].

The 674 ribs that were lighter than 135 grams came from 24 hoards and did not reveal a high enough peak to argue for standardization. Even when analyzed separately, the group had a maximum similarity index of 37.8%. The large variety in weights is evident also at the level of the hoards themselves (S1C Table in S1 Appendix). The ribs for the hoard of Temelín for instance, even when only compared to the other ribs from the same hoard, only reached a maximum similarity index of 38%, which is far lower than the numbers obtained for the hoards with heavy ribs.

Of the 208 axe blades dated to the EBA I, coming from 11 hoards, 44.2% weighed between 185 and 227 grams and would have been perceived as similar to an axe blade of 206 grams. While the similarity index is considerably lower than the results of rings and ribs, it is still higher than what one would expect in case of random data (Fig 4B). Furthermore, the dataset resulted in one peak only, with most weights falling within the same *perceptive category* of rings and heavy ribs. This conclusion was reinforced by the inability of the analysis to distinguish between EBA I axe blades, rings, and heavy ribs when combined into one dataset (Fig 3D). However, in the case of axe blades this weight standard only applied to some hoards, such as Sennwald-Salez, while hoards like Dermsdorf or Soběnice displayed a greater weight range, and others still, like Hindelwangen, followed a different pattern altogether (S1D Table in S1 Appendix).

The 401 axe blades from the EBA II showed even less homogeneity. The one larger peak found in the data, at 293 grams, with a maximum similarity index of 44.9%, seems to be the outcome of one hoard in particular, Gröbers-Bennewitz, and to a lesser degree the smaller hoard of Niederosterwitz. Taken together, they accounted for nearly half of the EBA II axe blades. When analyzed separately, the two hoards showed a remarkable similarity index of 80%. However, this standard cannot be related to the one found in the case of rings and ribs, since the two showed a difference of nearly 100 grams and would thus have been clearly discernible. Noteworthy too is that both hoards contain poorly made and even unfinished axe blades [40, 41]. The rest of the EBA II hoards did not fit the above pattern. Hoards like Soběchleby and Pilszcz revealed a large differentiation in terms of weight, and included axe blades that started from under 200 grams and went to over 400 grams. Other hoards, like those of Bresinchen and Carsdorf contained axe blades that overall came close to 200 grams, making them comparable to those from the EBA I. These two hoards contributed together to the smaller peak observed when analyzing all the EBA II axe blades. Both of these hoards also contained rings, suggesting a chronological and functional overlap.

## Discussion

There are two main directions for money definitions. They follow either a commodity theory (money as a means of exchange) or credit theory (money as a means of account) [42]. Essentially, the discussion between them revolves around the question whether the idea or the material expression came first, and is thus a matter of directional causality. Recently, this distinction has been challenged through findings that material practices scaffold mental processes, and cognition thus has a material dimension [43–45].

Most authors emphasise that exchange of commodity money is based on perceived alikeness [2, 5–7, 9, 42]. Commodity money displays rough similarities in terms of shape and weight, because of standardization, without necessarily following a strict metrological system.

Though archaeologists have no insight in the transactions that took place, there can be no doubt that at least the rings and ribs conform to the definition of commodity money. Our analysis revealed perceptible similarity in weight between rings, ribs, and a selection of axe blades from the Early Bronze Age of Central Europe. We take this to be evidence of intentional standardization that follows from lifting objects and estimating their weight by hand [25], attesting

for their use as commodity money. Standardization occurred around a weight of 195.5 grams, though this is not an absolute benchmark. It is unlikely that this *exact* weight was aimed for. Rather, the similarity in weight is the result of a rule of thumb corresponding to the material realities of casting metal in moulds and commodification. Moreover, in the absence of balances the shape of objects was needed to express weight through a simple 'same form same weight equation'. Employing psychophysics and the Weber fraction for weight of 0.1 we calculated that everything in the range from 176 to 217 grams would be perceived as equal in weight to 195.5 grams. Within this *perceptive category* we observe that ribs and axe blades form the opposite ends, with the rings at 195.5 in the middle. This argument is statistically supported by the observation that rings, ribs, and axe blades cannot be distinguished when combined in one dataset.

The co-occurrence of these objects in hoards, sometimes even tied together such as in the find from Wegliny [46] (Fig 5), points at their interchangeability. Their overlap in weight suggests that a conversion between rings and ribs, and axe blades was aimed for in specific cases. For the EBA I, one ring or rib corresponded to one axe blade, while during the EBA II, three rings or ribs were needed to make two axe blades. Since geographically axe blades were found at the outer edges of the area where rings and ribs circulated, we interpret them as local economic articulations of commodification through which the gap between commodity and prestige good economies was vaulted [10]. The choice of axe blades as commodity money seems a logical development from the observation that axes had long been valued, and are perhaps the single most important metal tool of prehistoric farmers [47].

Not all EBA rings, ribs, and axe blades were used as commodity money. What our model shows is that many conform to a standardized weight. Some were made for a different purpose which did not impose weight limitations that we can identify with this model, however. Axe blades from the Middle Bronze Age and Late Bronze Age showed no standardization (S1D Appendix, S1 Fig in S1 Appendix).

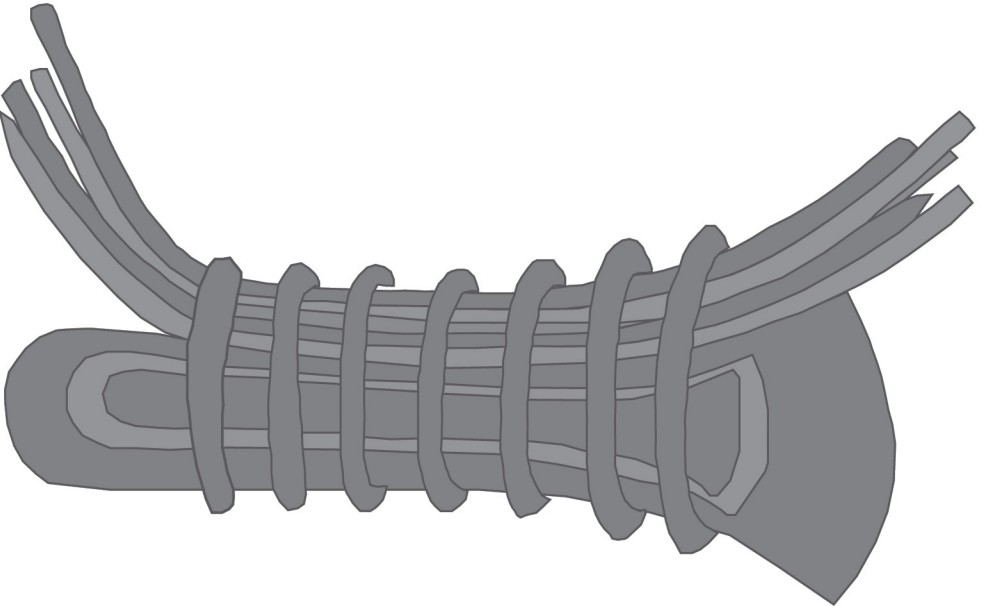

**Fig 5. Drawing of axe blade and ribs found bundled together in the hoard of Wegliny.** After Szpunar [46].

At the end of the Early Bronze Age rings and ribs disappear and trade starts to take place in both scrap metal and (parts of) casting cakes. For such a system to operate two developments need to have been completed. The practical development of a set of scales, and the cognitive development of a system of weighing through which to operate a balance. The earliest evidence of balance weights and balances in Western Europe dates to the Middle Bronze Age, and were likely used for gold given their sizes and weights [4, 48–50]. Around the same time there is a noticeable increase in the amount of bronzes being traded from the south to the north of Europe [51].

Weights are material-symbolic facts [52]. You first need to experience differences in weight and engage with these material realities before you can articulate them conceptually. Rather than seeing rings and ribs as the material representation of a conceptual system of weight, we argue that they helped to articulate such a conceptual system. The material medium—bronze —helped to move this conceptual innovation along [43, 53, 54] because it afforded an unprece-dented sameness between objects [55]. Moulds were the very first blue-prints [56] through which copies could be easily produced [57, 58]. Since human cognition is a dynamic system that includes mind, body and material forms [43, 44, 59, 60], thinking through these objects helped scaffold the cognitive framework that is needed for the development of a weight unit throughout the EBA [61].

From experience, people came to expect that rings weigh about the same (around 195 grams), resulting in a cognitive stereotype [62] of these rings and their weight. A cognitive stereotype is part of our cognitive toolbox and from this weight could be divorced from the actual physical rings, and thought of separately. Thanks to the particular affordances of bronze, equality in weight became a matter of concern and, following, a cognitive tool to think with, resulting in an abstract notion of weight. This is what allowed for a theoretical unit of weight to come into existence, which was needed to operate scales. Rings, ribs, and axe blades pro-duced in a serial fashion and having perceptible similar weight are the material roots of a cog-nitive system of weighing.

We suggest that producing perceptibly identical copies of rings, ribs, and axe blades, and their use as commodity money led to an increased recognition of weight similarities and the independent emergence of a system of weighing in Central Europe.

## Methods

### Psychophysics

The commonly used measure to express sensory acuity is the Weber fraction, which docu-ments the difference in stimulus strength that is just noticeable. The Weber fraction is calcu-lated with the formula DL/S = K. The DL (*Difference Limen)* is a statistical value that is defined as the just noticeable difference between the test stimulus and the initial stimulus (S) detected 50% of the time by observers based on a number of trials. The Weber fraction K is a constant [30, 31, 63].

Weight is the subjective perception of an object's mass when lifted, and thus effectively mass perception [33]. There is no fixed value for the Weber fraction of weight discrimination [64, 65]. This is because the value depends on the method of testing, the calculation of the threshold, and the range over which it is tested. Other key variables that influence weight per-ception are jiggling, skill, weight-illusions, and age [e.g. 66, 67]. Jiggling is common practice when assessing weight differences in real life because they improve our judgements [68]. We must assume that people in the Bronze Age were jiggling. Skill is important because practice influences weight discrimination [34]. We assume that the people trading in bronze were trained at discerning weights. Weight-illusions matter because our sensory modalities do not

operate independently from each other. Weight is mostly a tactile perception, but influenced by our visual perception of the weighted object. There is a size-weight illusion and a material-weight illusion. This is why in laboratory settings subjects typically are blinded, or given weights of equal size, but with a different mass. We assume that in the Bronze Age weight discrimination was partly also a matter of a visual size judgement. Taken these assumptions into account we use the generally-agreed Weber fraction of 0.1 for weight discrimination.

## Similarity index

To calculate similarity indexes, we rely on a comparison algorithm that integrates Weber fractions, which we implemented in R (version 3.5.1). The algorithm takes a parameter, in our case weight, and determines whether the two objects are perceptibly similar. We rely on the Weber fraction for weight to determine the threshold between similarity and difference.

The use of a similarity index has one main advantage over other methods such as histograms or mode calculations, even when the latter two are combined with Weber fractions: it allows for the identification of perceptive equivalence per individual objects. This feature makes the similarity index approach far more flexible and permits observations which would otherwise not be possible, such as average similarity per dataset (e.g. hoard, region) or number of objects in a dataset with similarity over a certain threshold.

In our approach, two objects are flagged as perceptibly similar when

$$W_Y - W_X \leq W_Y \times Wbf, \text{ if } W_X \leq W_Y$$

or when

$$W_X - W_Y < W_X \times Wbf, \text{ if } W_X > W_Y$$

$W_X$ and $W_Y$ refer to the weight of object X and object Y respectively, and Wbf refers to the Weber fraction for weight.

The weight of the heaviest object is employed to determine the threshold. This results in a similarity matrix, which records for each object whether it is perceptible similar or different from all other objects. In addition, each object receives a similarity index, which records the percentage of all compared objects to which object X is perceptibly similar. The outcome of this calculation is then smoothed and plotted on a two-dimensional graph displaying the similarity index of each object versus its weight. The shape of the graph indicates the presence or absence of preference for a particular weight standard, through the formation of peaks.

Since there is no definitive value for the Weber fraction for weight, a variety of values were tested, varying from 0.03 to 0.15. As expected, an increase in the Weber fraction leads to an overall increase in the similarity index of all objects. In addition, it induces a smoothing out of the distribution of similarity index values, particularly visible when displaying these values graphically. We settled on working with a Weber fraction of 0.1 in conjunction with previous research [32, 33], and the above discussed assumptions.

**Peaks and clustering.**   Peaks were determined both based on graphical observations and using the *peakpick* function from the *peakPick* statistical package [69]. The function proved to be accurate in identifying most of the peaks present in the data, fitting well with the graphical output.

Cluster analysis was used to confirm the presence of peaks and identify their separation thresholds. Clustering was performed based on the similarity matrix produced using Weber fractions. Two different clustering procedures were used: partitioning around medoids and hierarchical clustering. Partitioning around medoids is a more robust version of the k-means method [70] and was run using the *pam* function from the *cluster* statistical package [71]. The

function requires the user to stipulate the number of clusters that it should produce. We provided this data based on the graphical observation of the results. *pam* always assigns all cases to a cluster. Hierarchical clustering was performed using the *agnes* function [72, 73], also included in the statistical package *cluster*. Like most hierarchical clustering procedures, *agnes* results in a hierarchically organized dendrogram which requires further transformation, so-called cutting [72, 74], to determine the number of cluster and cluster membership. The dendrogram was cut using the *cutreeDynamic* function from the *dynamicTreeCut* package [75], which has the advantage of calculating group membership and the optimum number of groups. The function is also preferred because it does not necessarily assign all cases to a cluster, signaling out those that do not fit in the identified clusters.

**Random distribution test.** The different datasets were tested for random distribution. This involved first excluding any outliers from the dataset and then generating 999 sets of an equal number of randomly distributed data. We opted for a uniform distribution since it describes a scenario were all weights within the chosen interval would have the same likelihood to appear. Random data was generated with the *runif* function from the base package *stats* [76]. Following this, the random data was put through the similarity calculation. The results of the analysis, along with the random data scenario that gave the highest similarity index, were plotted against the original data for graphical comparison.

## Supporting information

**S1 Appendix. Supporting background and results.**
(DOCX)

**S2 Appendix. All rings, ribs and axe blades used for the analysis.**
(XLSX)

**S3 Appendix. Database sources and comments.**
(DOCX)

## Acknowledgments

We thank Majolie Lenerz- de Wilde for providing her original data and allowing us to digitalize this for publication. Further help in collecting data came from Mario Küßner (Landesambt für Denkmalpflege und Archäologie Weimar), Florian Klimscha (Landesmuseum Hannover), Ronny Leder and Jörg Frase (Naturkundemuseum Leipzig), Daniela Messerschmidt (Regionalgeschichtliche Sammlungen der Lutherstadt Eisleben), Kerstin Hoffman and Knut Rassmann (Römisch-Germanische Kommission, Frankfurt), Heiner Schwarzberg (Archäologisches Staatssammlung München), Uwe Reuter and Anja Kaltofen (Landesamt für Archäologie Sachsen), Christian Later (Bayerisches Landesamt für Denkmalpflege), Veit Dresely (Landesamt für Denkmalpflege und Archäologie Sachsen-Anhalt), Günther Moosbauer (Gäubodenmuseum Straubing), Franz Pieler (Landessammlungen Niederösterreich), and Bernard Heeb (Museum für Vor- und Frühgeschichte, Berlin). We received help from five enthusiastic students compiling the database: Mignonne Lenoir, Jan Dekker, Jesper de Munnik, Mika van Eldijk, and Florian Helmecke. We further thank David Fontijn, Quentin Bourgeois, Marie Soressi, and Marianne Mödlinger for their constructive feedback, in addition to our anonymous reviewers. Part of this research was written during a research stay at the Max Planck Institute for the History of Science where one of the authors was a visiting postdoctoral fellow (MHGK).

## Author Contributions

**Conceptualization:** Maikel H. G. Kuijpers, Cătălin N. Popa.

**Data curation:** Maikel H. G. Kuijpers, Cătălin N. Popa.

**Formal analysis:** Maikel H. G. Kuijpers, Cătălin N. Popa.

**Investigation:** Maikel H. G. Kuijpers, Cătălin N. Popa.

**Methodology:** Maikel H. G. Kuijpers, Cătălin N. Popa.

**Validation:** Maikel H. G. Kuijpers, Cătălin N. Popa.

**Visualization:** Maikel H. G. Kuijpers, Cătălin N. Popa.

**Writing – original draft:** Maikel H. G. Kuijpers, Cătălin N. Popa.

**Writing – review & editing:** Maikel H. G. Kuijpers, Cătălin N. Popa.

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
