## [Decision Letter · Decision Letter 0]

5 Aug 2020

PONE-D-20-19442

The origins of money: Calculation of similarity indexes demonstrates the earliest development of commodity money in prehistoric Central Europe

PLOS ONE

Dear Dr. Kuijpers,

Thank you for submitting your manuscript to PLOS ONE. After careful consideration, we feel that it has merit but does not fully meet PLOS ONE’s publication criteria as it currently stands. Therefore, we invite you to submit a revised version of the manuscript that addresses the points raised during the review process.

All comments have to be addressed before re-submission.

We look forward to receiving your revised manuscript.

Kind regards,

Peter F. Biehl, PhD

Academic Editor

PLOS ONE

Journal Requirements:

2.In your manuscript, please provide additional information regarding the specimens used in your study. Ensure that you have reported specimen numbers and complete repository information, including museum name and geographic location.

For more information on PLOS ONE's requirements for paleontology and archaeology research, see https://journals.plos.org/plosone/s/submission-guidelines#loc-paleontology-and-archaeology-research.

3.We note that [Figure(s) 1] in your submission contain [map/satellite] images which may be copyrighted. All PLOS content is published under the Creative Commons Attribution License (CC BY 4.0), which means that the manuscript, images, and Supporting Information files will be freely available online, and any third party is permitted to access, download, copy, distribute, and use these materials in any way, even commercially, with proper attribution. For these reasons, we cannot publish previously copyrighted maps or satellite images created using proprietary data, such as Google software (Google Maps, Street View, and Earth). For more information, see our copyright guidelines: http://journals.plos.org/plosone/s/licenses-and-copyright.

1.    You may seek permission from the original copyright holder of Figure(s) [1] to publish the content specifically under the CC BY 4.0 license. 

Additional Editor Comments (if provided):

All comments have to be addressed before re-submission.

Reviewers' comments:

Reviewer's Responses to Questions

**Comments to the Author**

1. Is the manuscript technically sound, and do the data support the conclusions?

Reviewer #1: Partly

Reviewer #2: Yes

2. Has the statistical analysis been performed appropriately and rigorously? 

Reviewer #1: Yes

Reviewer #2: I Don't Know

3. Have the authors made all data underlying the findings in their manuscript fully available?

Reviewer #1: Yes

Reviewer #2: Yes

4. Is the manuscript presented in an intelligible fashion and written in standard English?

Reviewer #1: Yes

Reviewer #2: Yes

5. Review Comments to the Author

Reviewer #1: This is an interesting paper that draws significant conclusions for the history of metrology and for Bronze Age Europe based on an original analysis of data from hoards of rings, ribs and axes.

I have a few comments and queries. It's because of my queries below that I've answered 'partly' toQ1.

l 79 Weighing was thus a qualitative measurement based on comparative sensory.

Better to say 'would have been' rather than 'was', which already implies that weight mattered.

l 95-6 As a general rule, the Weber fraction for weight discrimination is 0.1

Give a numerical example for clarity

l 101-103 The perceptive equivalence is expressed though the calculation of a similarity index (SI), which gives the percentage of objects from a dataset that are perceptibly indistinguishable from a tested object.

Would help to give a worked example in the main text and also to explain why the similarity index method is necessary, as opposed to simply plotting the distribution, identifying the mode and the Weber fraction range around it and saying what proportion are indistinguishable.

l 117 similarity index of 58.5%

Hard to get a sense of how these indices relate to the Weber fraction. Spell out more?

l 132 randomly distributed data

Doesn't say the form of distribution used to randomly generate the data or why. Uniform? Normal? Again a more detailed description with a worked example would help.

Reviewer #2: This very interesting paper reinforces what many of us have long suspected: that estimation of quantities (including weight) was a matter of perception rather than precise measurement. I am not familiar with the Weber fraction but from your presentation it appears to have been employed usefully.

Four other discussions of rings and ribs, not included in the references, might be included:

Menke, M. 1978–9. Studien zu den frühbronzezeitlichen Metalldepots Bayerns, Jahresbericht der bayerischen

Bodendenkmalpflege 19–20, 5–305.

Eckel, F. 1992. Studien zur Form- und Materialtypologie von Spangenbarren und Ösenringbarren, zugleich ein Beitrag zur

Frage der Relation zwischen Kupferlagerstätten, Halbzeugproduktion und Fertigwarenhandel. Saarbrücker Beiträge zur

Altertumskunde 54. Bonn: Habelt.

Bath-Bílková, B. 1973. K problému puvodu hriven, Památky Archeologické 64, 24–41.

Harding, A. 1983. The Bronze Age in central and eastern Europe: advances and prospects, Advances in World Archaeology

2, 1–50.

6. PLOS authors have the option to publish the peer review history of their article (what does this mean?). If published, this will include your full peer review and any attached files.

Reviewer #1: No

Reviewer #2: No

---

## [Author Response · Author response to Decision Letter 0]

18 Sep 2020

In point 2 additional information regarding the specimens was asked for, specifically repository information. We have added museum and geographical information. For the axes also individual inventory numbers from the respective museums were added, and the number used in the main publication. We could not do the same for the rings and ribs. Because of the sheer number of near-identical objects in hoards (up to 500 pieces) these finds are often stored under the hoard name, rather than given individual inventory numbers. 

Additionally, we have merged the two excel sheets containing the data in one appendix (excel) using tabs. Three additional tabs were added. One with the full names of the museums in their respective language and their location. And two tabs that list the ring and axe-hoards, including coordinates, repository information, and literature. 

Bibliography of the database and the permit disclosure has been added in appendix 3. 

Point 3 had to do with figure 1. This GIS map of the country contours comes from Natural Earth. We have added this in the figure description. 

Reviewer comments:

Reviewer 1: 

l 79 Weighing was thus a qualitative measurement based on comparative sensory.

Better to say 'would have been' rather than 'was', which already implies that weight mattered.

While we initially made the change suggested by the author, we decided to go back to the use of “was” since “would have been” added uncertainty where there is none. In the absence of a weighing apparatus, comparing through one’s senses was the only possibility.

l 95-6 As a general rule, the Weber fraction for weight discrimination is 0.1

Give a numerical example for clarity

We have added a numerical example for clarity.

l 101-103 The perceptive equivalence is expressed though the calculation of a similarity index (SI), which gives the percentage of objects from a dataset that are perceptibly indistinguishable from a tested object. Would help to give a worked example in the main text and also to explain why the similarity index method is necessary, as opposed to simply plotting the distribution, identifying the mode and the Weber fraction range around it and saying what proportion are indistinguishable.

A small paragraph has been added describing the advantage of using similarity indexes over other methods, such as those proposed by the reviewer. The paragraph was added in the Methods section (lines 302-307), since it would have been unfitting to add such a discussion in the Data section to which the reviewer refers. We added a reference here to guide the reader: (methods)(line 106).

l 120 similarity index of 58.5%. Hard to get a sense of how these indices relate to the Weber fraction. Spell out more?

We added a sentence explaining how this relates to the Weber fraction for weight.

l 135 randomly distributed data

Doesn't say the form of distribution used to randomly generate the data or why. Uniform? Normal? Again a more detailed description with a worked example would help.

We added a sentence mentioning that uniform distribution was used, and a short argument for doing so. The sentence was added in the Random Distribution section (lines 346-348) since it applies to all instances were random data was generated, rather to the one instance the reviewer points at on line 132. We do not find that a more detailed discussion would significantly increase clarity. 

Reviewer 2 

Reviewer two had no questions or remarks besides suggesting some additional literature. Of the four mentioned, Menke (1978-9) was already mentioned in the appendix. We have read and added the other three in the appendix (line 12) as they indeed provided some additional archaeological background.

---

## [Editor Report · Decision Letter 1]

28 Sep 2020

The origins of money: Calculation of similarity indexes demonstrates the earliest development of commodity money in prehistoric Central Europe

PONE-D-20-19442R1

Dear Dr. Kuijpers,

We’re pleased to inform you that your manuscript has been judged scientifically suitable for publication and will be formally accepted for publication once it meets all outstanding technical requirements.

Kind regards,

Peter F. Biehl, PhD

Academic Editor

PLOS ONE
---

## [Editor Report · Acceptance letter]

7 Dec 2020

PONE-D-20-19442R1 

The origins of money: Calculation of similarity indexes demonstrates the earliest development of commodity money in prehistoric Central Europe 

Dear Dr. Kuijpers:

I'm pleased to inform you that your manuscript has been deemed suitable for publication in PLOS ONE. Congratulations! Your manuscript is now with our production department. 

Kind regards, 

on behalf of

Dr. Peter F. Biehl 

Academic Editor

PLOS ONE